# ADP is the dominant controller of AMP-activated protein kinase activity dynamics in skeletal muscle during exercise

**Ian F. Coccimiglio** [ID]**, David C. Clarke** [ID]*

Department of Biomedical Physiology and Kinesiology and Centre for Cell Biology, Development and Disease, Simon Fraser University, Burnaby, Canada

* dcclarke@sfu.ca

**Data Availability Statement:** All relevant data are within the manuscript and its Supporting Information files.

**Funding:** This work was supported by a Natural Sciences and Engineering Research Council of

## Abstract

Exercise training elicits profound metabolic adaptations in skeletal muscle cells. A key molecule in coordinating these adaptations is AMP-activated protein kinase (AMPK), whose activity increases in response to cellular energy demand. AMPK activity dynamics are primarily controlled by the adenine nucleotides ADP and AMP, but how each contributes to its control in skeletal muscle during exercise is unclear. We developed and validated a mathematical model of AMPK signaling dynamics, and then applied global parameter sensitivity analyses with data-informed constraints to predict that AMPK activity dynamics are determined principally by ADP and not AMP. We then used the model to predict the effects of two additional direct-binding activators of AMPK, ZMP and Compound 991, further validating the model and demonstrating its applicability to understanding AMPK pharmacology. The relative effects of direct-binding activators can be understood in terms of four properties, namely their concentrations, binding affinities for AMPK, abilities to enhance AMPK phosphorylation, and the magnitudes of their allosteric activation of AMPK. Despite AMP's favorable values in three of these four properties, ADP is the dominant controller of AMPK activity dynamics in skeletal muscle during exercise by virtue of its higher concentration compared to that of AMP.

## Author summary

During exercise, the enzyme "AMP-activated protein kinase" (AMPK) detects the disrupted cellular energy state by binding to the adenine nucleotides ATP, ADP, and AMP, which are the major chemical energy carriers of the cell. How the adenine nucleotides interact to control AMPK signaling dynamics is incompletely understood. In this study, we used mathematical modeling to investigate the control of AMPK signaling by the adenine nucleotides in skeletal muscle during exercise. We simulated the model many times with randomly generated parameter sets sampled from experimentally determined ranges of plausible values. Ultimately the parameters affect four key properties of an AMPK activator, namely its concentration, the tightness with which it binds to AMPK, its ability to activate AMPK by promoting its phosphorylation, and its ability to activate AMPK

Canada (https://www.nserc-crsng.gc.ca/index_eng.asp) Discovery Grant to D.C.C. (RGPIN 06004-2014). The funders had no role in study design, data collection and analysis, decision to publish, or preparation of the manuscript.

through allostery. We found that ADP is the dominant controller of AMPK activity, instead of AMP, due to its high concentration relative to that of AMP. We also modeled AMPK activity in response to drugs that activate it, which further demonstrated the validity and applicability of the model. Overall, our research enhances understanding of AMPK action during exercise and could inform the development of drugs that target AMPK.

## Introduction

Exercise training induces profound fitness-promoting adaptations in skeletal muscle cells [1]. How the "stress" of exercise is sensed by cells and translated into adaptations is unclear. It is known that a protein signaling network exists within skeletal muscle cells, the components of which are directly activated by the diverse biochemical and biophysical stressors of exercise [2]. The proteins within the network transmit signals in the form of altered rates of biochemical reactions such as phosphorylation. The information is transmitted to effectors such as transcription factors in the nucleus, whose altered activities increase the expression of fitness-promoting genes. What remains to be determined is how different durations and intensities of exercise are encoded by the network and translated into specific adaptations.

AMP-activated protein kinase (AMPK) is a prototypical exercise-responsive signaling protein. AMPK exists as a complex composed of three subunits, α, β, and γ, each of which has two or three isoforms (namely α1 and α2, β1 and β2, and γ1, γ2, and γ3). Any combination of the α, β, and γ isoforms are possible but in human skeletal muscle the predominant complexes are α1β2γ1 (15%), α2β2γ1 (65%), α2β2γ3 (20%) [3]. In mice, some muscles also express β1-containing complexes [3]. Mice featuring muscle-specific deletions of both AMPK β-subunits exhibit reduced mitochondria and exercise tolerance, thus demonstrating the importance of AMPK in facilitating muscle adaptations to physical activity [4]. In humans, AMPK signaling increases in response to diverse types of exercise [5,6]. AMPK substrates include transcription factors such as cAMP-response element binding protein (CREB) and coactivators such as peroxisomal proliferator-activated receptor-γ co-activator-1α (PGC-1α) [7]. The phosphorylation of these transcriptional regulators by AMPK stimulates their abilities to enhance the expression of metabolic proteins such as hexokinase II [8] and mitochondrial proteins [9]. Skeletal-muscle AMPK function is important for promoting whole-body metabolic health and has motivated the development of "exercise-mimetic" drug cocktails, a core component of which are small-molecule AMPK activators [10,11]. It is therefore important to better understand AMPK activation in skeletal muscle in response to exercise and pharmacological activators.

AMPK activity is dynamically controlled by phosphorylation and allostery, both of which are prompted by changes to the bioavailable ("free") concentrations of the adenine nucleotides (AXPs), i.e., adenosine tri-, di-, and monophosphate (ATP, ADP, and AMP, respectively) [12]. AMPK is phosphorylated at a threonine residue (Thr172) within the activation loop of the α-subunit [13]. Several kinases and phosphatases control the phosphorylation state of this site *in vivo*. The principal kinases include liver kinase B1 (LKB1) and calmodulin-dependent protein kinase kinase β (CAMKKβ) [14], and the main phosphatases include phosphoprotein phosphatases 1 and 2A (PP1, PP2A), and metal-dependent protein phosphatase 2C (PP2C) [15–17]. ADP and AMP inhibit phospho-AMPK Thr172 dephosphorylation [18,19] and promote AMPK Thr172 phosphorylation by LKB1 and CaMKKβ by enhancing AMPK's ability to serve as a substrate for these upstream kinases [14,20–22]. ATP inhibits AMPK phosphorylation by competing with ADP and AMP for binding to AMPK and by stabilizing it in a conformation

that favors its dephosphorylation [17]. AMP allosterically activates AMPK catalytic activity by 1.5- to 13-fold, depending on the estimate, by binding to cystathionine-β-synthase (CBS) domains within the γ subunit [14,19,22]. How these factors interact to control overall AMPK activity is poorly understood.

Two principal hypotheses have been proposed to explain AXP-mediated control of AMPK activity *in vivo*. One hypothesis posits that AMP (or its ratio with ATP) is the primary controller of AMPK [23,24]. Three lines of argument support this hypothesis. First, AMP is proposed to act as a more sensitive controller of AMPK activity than ADP for a given amount of ATP breakdown because the adenylate kinase reaction involves ATP and AMP interconverting into two ADP, such that the AMP:ATP ratio would vary as the square of the ADP:ATP ratio [25]. Second, AMP enhances the phosphorylation of AMPK by LKB1 and by CaMKKβ by up to four fold [14,21,22], whereas ADP enhances this phosphorylation by only ~1.5-fold [14,20]. Third, only AMP allosterically activates AMPK [14,19,22], the degree to which depends on the γ-subunit isoform within the AMPK complex, with γ2-containing complexes being strongly activated and γ3-containing complexes exhibiting little to no activation [14,26]. Overall, AMP is more potent on a per-molar basis than ADP in activating AMPK [22,24], which ultimately reflects a higher *per-molecule activation potency*, i.e., the degree of activation per AMPK molecule upon the binding of activator.

An alternative hypothesis is that ADP is the primary controller of AMPK activity, which is based on two main arguments. First, the concentration of free ADP is orders of magnitude higher than that of free AMP (50–200 μM versus 0.5–5 μM) [17,18,20,27,28]. The disparity in the intracellular concentrations of ADP and AMP is a manifestation of the adenylate kinase reaction operating near equilibrium and is further promoted by the AMP deaminase reaction, in which AMP is degraded into inosine monophosphate (IMP) and ammonia [20,29]. This reaction is particularly discernible during maximal-intensity sprint exercise, because intramuscular IMP levels increase from undetectable levels at rest to ~20 μmol·g dry mass$^{-1}$ after exercise [30]. Second, ADP and AMP both bind to the AMPK γ-subunit nucleotide-binding site 3 with roughly similar affinities [18]. Data from human exercise studies show that the concentration of ADP exceeds its dissociation constant ($K_D$) of binding to site 3 on the γ domain, whereas the concentration of AMP does not, thus implying that ADP outcompetes AMP for binding to that site [27]. In accordance, ADP levels and AMPK activities increase in parallel as a function of exercise intensity [27]. Together, the comparable affinities of binding and the higher ADP concentration may enable it to serve as the primary controller of AMPK despite its lesser per-molecule activation potency compared to AMP.

Despite the rich *in vitro* and *in vivo* data available regarding AMPK activity control, these hypotheses await definitive support or refutation because of the complex and quantitative nature of AMPK control. Complexities include the dynamic variation of AXP concentration with exercise intensity, the effects of the comparable yet quantitatively different binding affinities of the AXPs towards the γ isoforms [31,32], and the different per-molecule activation potencies of the AXPs towards AMPK. These factors are challenging to parse experimentally. Accordingly, a more complete understanding of AMPK activation *in vivo* requires the study of these mechanisms using methods that account for these complexities. Enhanced understanding of AMPK activation would inform both exercise training biology and AMPK pharmacology.

Mathematical modeling is a powerful tool for investigating the dynamics of complex biochemical networks. To date, physicochemical models featuring AMPK have studied its control by insulin and mTORC1 [33], as well as AMPK's roles in controlling the balance of protein translation and autophagy [34], in maintaining circadian metabolic function of the liver [35], and in cancers such as glioblastoma [36]. However, none of these studies focused on AXP-mediated control mechanisms of AMPK, none of the models featured ADP-mediated control of AMPK, and none featured skeletal muscle as the tissue of interest. Given that ATP turnover

can increase 100-fold over rest in muscle during intense exercise [37], a dedicated model of AMPK control in skeletal muscle is warranted.

In this study, we used mathematical modeling to investigate the control of AMPK activity dynamics by AXPs in skeletal muscle in response to exercise. Our model combines a published model of AXP dynamics in contracting skeletal muscle with a new model of AMPK signaling that integrates all known AXP-mediated control mechanisms. We used the model to test the hypothesis that ADP is the dominant controller of AMPK activity in skeletal muscle during exercise. Specifically, we performed global parameter sensitivity analyses guided by a framework in which AMPK activation was expressed in terms of the activator concentration, binding affinity, phosphorylation enhancement, and allosteric activation. The model generalized to direct-binding AMPK activators, thus extending the its applicability to understanding AMPK pharmacology.

## Methods

### Model description

A schematic diagram of the model topology is presented in Fig 1. The model is briefly described here, with comprehensive details and justification provided in the Supplementary

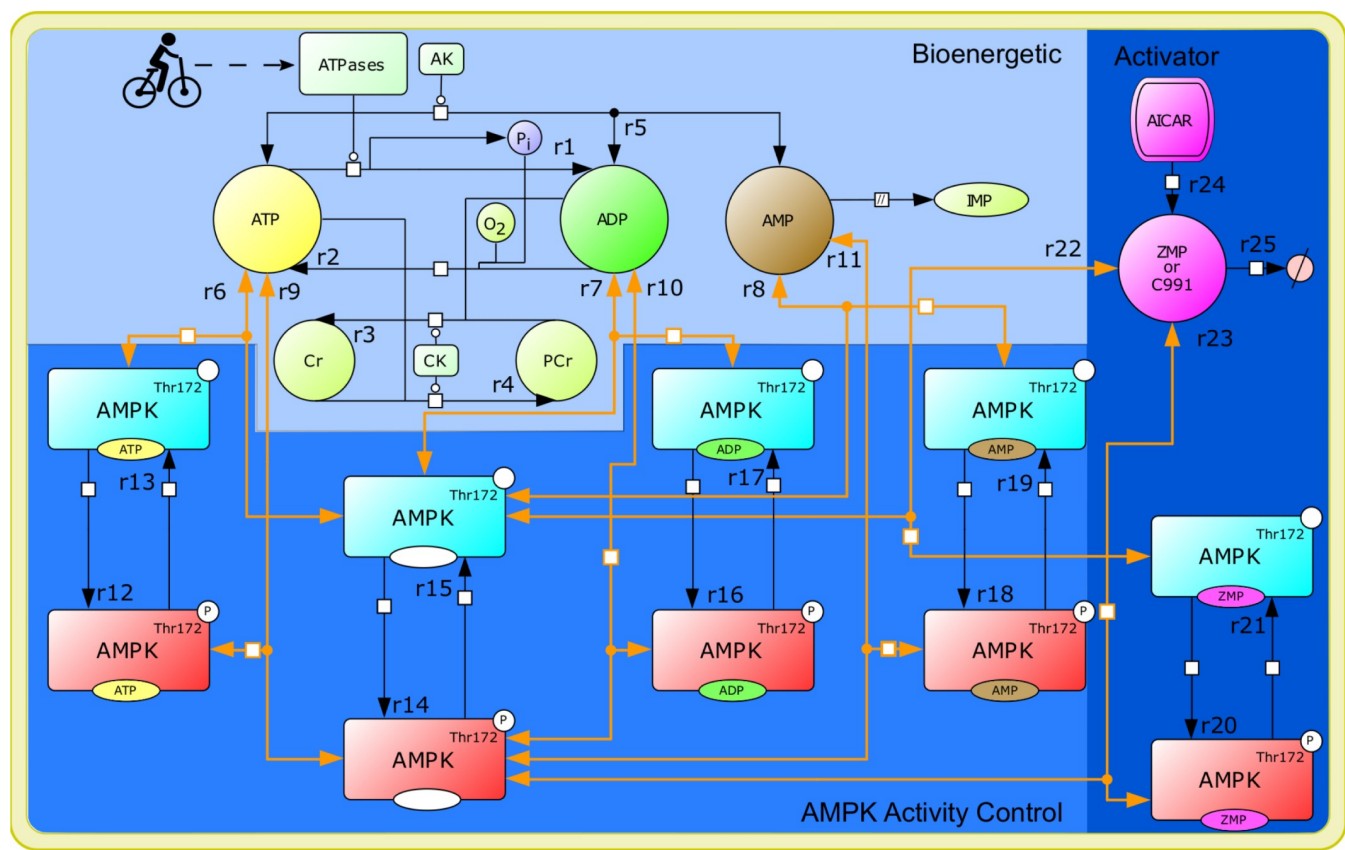

**Fig 1. AMPK model reaction diagram.** Biochemical reactions are denoted by arrows and are labeled with a reaction number (r1, r2, etc.). Reversible reactions are indicated by double-sided arrows. Orange arrows depict nucleotide-binding reactions, and black arrows depict enzyme-catalyzed synthesis and degradation reactions. Species colored cyan represent unphosphorylated AMPK and species colored red represent phospho-AMPK. To simulate exercise, we changed the rate of the ATPase reaction (r1), which serves as the input to the model. The three modules of the model are distinguished by shading with the bioenergetic module shaded light blue, the AMPK regulatory module medium blue, and the activator module darker blue. The activator module is used to simulate the effects of two activators, ZMP and Compound 991 (C991). When simulating C991, only reactions r22 and r23, which represent reversible binding of C991 to AMPK and phospho-AMPK, are simulated in the activator module. When simulating the effects of ZMP, the module features r22 and r23 as well as r24 and r25, which represents the interconversion of 5-aminoimidazole-4-carboxamide ribonucleoside (AICAR), the precursor molecule, to ZMP (r24) and ZMP degradation (r25). Activator-specific parameter values are used for r22 and r23.

Methods (S1 Text, "Model construction: System Definition" and "Model construction: Topology" subsections). The system is defined as a muscle fiber with average contractile, metabolic, and fatigability properties contracting within a primary agonist muscle during cycling exercise. In the spectrum from slow to fast-twitch fibers, the fiber would most closely resemble a type-IIa fiber [38]. The model is expressed as a system of ordinary differential equations that describe the rates of change of the represented molecular species. The model consists of three modules: 1) bioenergetic, 2) AMPK regulatory, and 3) pharmacological activator (PA). The bioenergetic module simulates AXP dynamics in response to exercise and is based on published models [39,40]. The inputs to the model were the parameters describing the rates of ATP hydrolysis under different conditions ($k_{rest}$, $k_{stim}$, $k_{post}$; Table D in S1 Text). These parameters were set to values that led the predicted phosphocreatine concentrations to match measured values. The outputs of the model were the predicted phospho-AMPK (p-AMPK) levels and the activities of the various AMPK species. The activity of each molecular species was calculated as the product of its concentration and corresponding $k_{cat}$ parameter. Model species and initial conditions, reaction rate equations, ordinary differential equations, and parameter values are presented in Tables A, B, C, and D in S1 Text, respectively. Allosteric activation potencies, simulation-specific parameter values, and activator properties are presented in Tables E, F, and G in S1 Text, respectively.

## Model calibration

The model parameter values (Tables D, E, F, and G in S1 Text) were calibrated by using estimates from published literature, biochemical calculations, and manual adjustment to fit the model to data from exercising humans. The calibration is described in detail in the Supplementary Methods (S1 Text, "Model Calibration" subsection).

## Model validation

To externally validate the model, the predictions of the calibrated model were compared against two independent datasets: one collected from a continuous exercise protocol [41] and the other from a sprint-interval exercise protocol [6]. To simulate exercise intensity, we modified the $k_{rest}$ and $k_{stim}$ values until the modeled PCr levels matched the experimentally observed values at rest and during exercise. The values for all other model parameters and initial conditions were those of the calibrated model.

## Global parameter sensitivity analyses

We performed global sensitivity analyses to systematically explore the parameter determinants of AXP dominance in controlling AMPK activity. Specifically, we employed Multi-Parametric Sensitivity Analysis (MPSA) [42] as follows. First, we generated 50,000 parameter sets by selecting values from logarithmically scaled ranges according to the Latin hypercube sampling method [43]. Latin hypercube designs efficiently ensure even sampling of all parameters across their sampling ranges [43]. Accordingly, parameter sensitivities are assessed in the context of the full range of values for the other parameters, such that interaction effects between parameters, if they exist, should be detectable.

We performed three MPSA, denoted "unconstrained", "α1β2γ1 $K_D$", and "α2β2γ3 $K_D$". For each, the sampling ranges for kinetic parameters associated with the bioenergetic module were centered around their published estimates [39,40] (S1 Spreadsheet, "Parameter Inputs" worksheet). Those for the AMPK regulatory module were set in the unconstrained MPSA based on the limits of biophysical plausibility [44]. In the other two MPSA, the sampling ranges were

±10% around the measured $K_D$ values for two AMPK isoforms, α1β2γ1 and α2β2γ3 (S1 Spreadsheet, "Binding kinetics" worksheet).

The resulting models were then simulated, and the outputs used to compute classification metrics for analysis. Models were retained for analysis if their simulated phospho-AMPK time courses exhibited acceptable qualitative trends based on the criteria stated in the Supplementary Methods (S1 Text, "Multi-parametric sensitivity analysis: Criteria for acceptable models" subsection). For each acceptable model, the degrees to which ADP and AMP each contributed to the control of AMPK activity were assessed by numerically computing the time integrals of ADP-bound phospho-AMPK (ADP-p-AMPK) and AMP-bound phospho-AMPK (AMP-p-AMPK), followed by computing the *fraction of ADP-mediated control* as follows:

$$fraction\ ADP\ control = \frac{[ADP-p-AMPK]_{AUC}}{[ADP-p-AMPK]_{AUC} + [AMP-p-AMPK]_{AUC}} \quad (1)$$

Models were classified as "AMP-dominant" or "ADP-dominant" if this fraction was $< 0.5$ or $> 0.5$, respectively. The dependencies of AXP dominance on individual kinetic parameters were assessed by computing Kolmogorov-Smirnov statistics. In addition, relationships between the parameter values and the *fraction ADP control* were assessed by computing Spearman correlation coefficients. Parameter effects were visualized by plotting boxplots of the parameter values from AMP- and ADP-dominant models. To visually accentuate any differences, the values from parameter sets leading to *fraction ADP control* $> 0.8$ ("ADP-dominant") or $< 0.2$ ("AMP-dominant") were used for constructing the boxplots.

AXP dominance is expected to be a function of interactions between multiple parameters rather than single isolated parameters. Because the number of model parameters was prohibitively large to tractably apply standard statistical approaches, we derived a framework from which we specified an analysis plan (see Supplementary Methods in S1 Text, "Multi-parametric sensitivity analysis: Framework for analyzing AMPK activation" subsection). Equation G (S1 Text) compactly expresses the five factors that determine total AMPK activity, namely the concentration of activator, the binding affinity of the activator for AMPK, the concentration of phospho-AMPK and its activity, and the allosteric activation of AMPK provided by the activator. Equation I (S1 Text) then provides the basis for comparing the potencies of two activators in terms of intuitive groupings of model parameters that determine the listed factors. For AXP-(p)-AMPK binding parameters, boxplots were constructed for individual forward and reverse binding rate constants ($k_f$ and $k_r$), the $K_D$ values, and the ratios of $K_D$ values. For activator concentrations, the maximum-predicted AMP and ADP concentrations were plotted, as was the ratio between these values. The parameter values representing allosteric activation of AMPK by AMP were analyzed in isolation because ADP does not allosterically activate AMPK. The phosphorylation enhancement was assessed by examining individual $V_{max}$ and $K_M$ values for the AMPK kinase and phosphatase, followed by $V_{max}/K_M$ values, which provides an index for the overall parameter effects in the Michaelis-Menten equation, and then by compound ratios of $V_{max}/K_M$ values for the kinases and phosphatases.

## Software and numerical algorithms

CellDesigner 4.4.2 [45] was used to create the model reaction diagram (Fig 1). MATLAB version r2019a (9.6.0.1099231) was used to perform all analyses and calculations. Latin Hypercube sampling was done using the 'lhsdesign' function in MATLAB, the areas under the curve (AUC) were calculated using the 'cumtrapz' function, and the model was numerically integrated using the 'ode23s' algorithm using default tolerances. We also solved the model using alternative solvers and confirmed that our choice of solver did not affect the model results.

Code to run the model and reproduce Figures 2 and 3 is included as supplementary files (S1 Code).

## Results

### Bioenergetic module simulation of AXP dynamics

The AMPK model predicts AMPK dynamics resulting from AXP dynamics during exercise. The experimental data for AMPK signaling against which our model was compared came from the vastus lateralis of humans during cycling exercise. Hence, we first verified whether the bioenergetic module, which was based on a model of metabolism in contracting forearm skeletal muscle [39], could adequately simulate AXP dynamics in this different mode of exercise [5]. The concentrations of PCr, inorganic phosphate ($P_i$), and the AXPs were predicted to achieve steady states at ~10 min following the onset of exercise (Fig 2). The PCr concentration decreased from 23 mM to 17 mM and the $P_i$ concentration increased from 2.6 mM to 8.7 mM (Fig 2A). ATP concentration was essentially preserved at its resting levels of 7.5 mM, which matched the data well (Fig 2B). ADP and AMP increased by 1.8-fold (49 μM to 89 μM) and 3.4-fold (0.43 μM to 1.4 μM), respectively, which also matched well their experimentally observed values (Fig 2B). These results demonstrate that the bioenergetic module satisfactorily explained AXP dynamics in vastus lateralis during cycling exercise.

### Model calibration results and predicted AMPK dynamics during continuous submaximal exercise

Next, we calibrated the model to AMPK signaling data and evaluated the predicted AMPK activity dynamics. Initial simulations revealed that the model with the pre-calibration parameter values caused the phospho-AMPK levels to rise and fall unrealistically fast. Guided by the local parameter sensitivity analysis, we found that this discrepancy could be reduced by lowering the kinase and phosphatase parameters from their pre-calibration values (40–47, Table D in S1 Text). The kinetic parameters pertaining to AXP-AMPK binding were also adjusted (Table D in S1 Text). $K_{eqADK}$ was decreased to better reflect human tissue in which the intracellular $Mg^{2+}$ differs from the reported data (Table D in S1 Text). Collectively, these changes slowed the rate of AMPK phosphorylation and dephosphorylation and increased free AMP levels, such that the model predictions better matched experimental measurements.

The calibrated model predicted the following features of AMPK dynamics in skeletal muscle during exercise. First, the concentration of phospho-AMPK increased from 34% to 45% of the total amount of AMPK (Fig 2C). The total kinase activity of phospho-AMPK increased ~30% during exercise, 93% and 4% of which was attributable to the ADP- and AMP-bound forms of phospho-AMPK, respectively (Fig 2D). On average, the experimental data showed that AMPK activity increased ~40% after 30-minutes of exercise [5]. To better match the model output to the data, we tried increasing the $V_{max}$ values for AMPK phosphorylation (parameters 40–43, Table D in S1 Text). While this strategy led to better fits of the activity data, it also caused the rise in phospho-AMPK levels to be excessively rapid, such that the resulting fit is a compromise.

### Model validation and AMPK dynamics during sprint-interval exercise

Next, we validated the calibrated model against the data of Nielsen et al. [41] and Gibala et al. [6]. The Nielsen study featured sedentary and trained humans continuously exercising for 20 min at an intensity of 80% of $\dot{V}O_{2max}$, before and after which PCr concentrations, relative

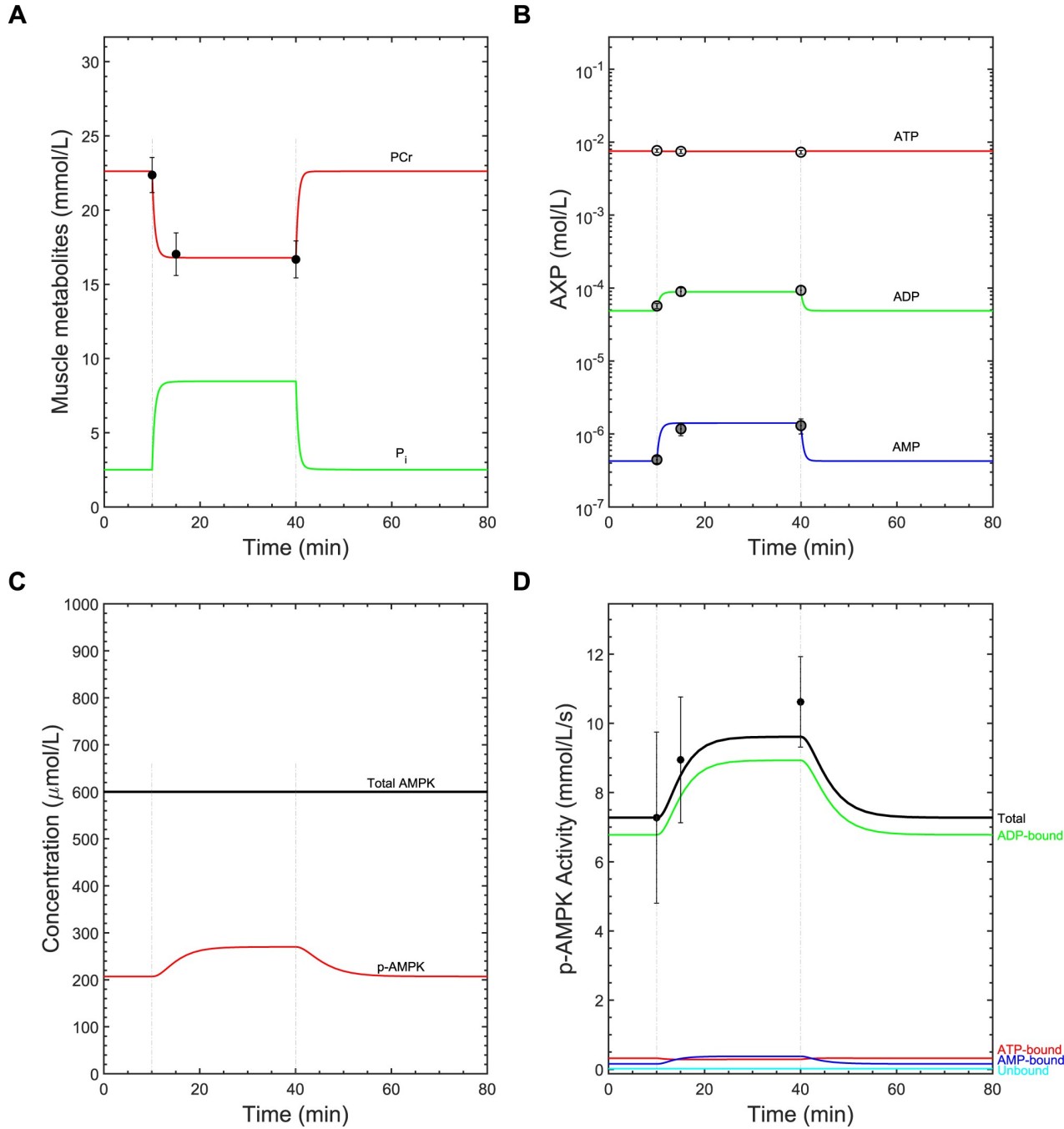

**Fig 2. Model calibration results.** The model was initially calibrated to data collected from a 30-min submaximal-intensity cycling exercise protocol [5]. For all panels, the measured data are presented as means ± standard error, if available, and the hatched vertical lines indicate the exercise initiation and cessation times. A) Time courses of phosphocreatine (PCr) and inorganic phosphate ($P_i$) concentrations. The filled circles represent measured PCr concentrations. B) Semi-logarithmic plot showing the time courses of AXP concentrations in response to exercise. The open points, light gray points, and dark gray points represent the measured ATP, ADP, and AMP concentrations, respectively. C) Time courses of the total and phospho-AMPK (p-AMPK) levels. D) Time courses of the simulated kinase activities of each AXP-bound phospho-AMPK (AXP-p-AMPK) complex. Filled circles represent the measured mean relative change in total AMPK activity between rest and exercise.

phospho-AMPK levels, and AMPK activities were measured in muscle biopsies. The model reproduced the relative constancy of ATP concentrations during exercise (Fig 3B and 3F),

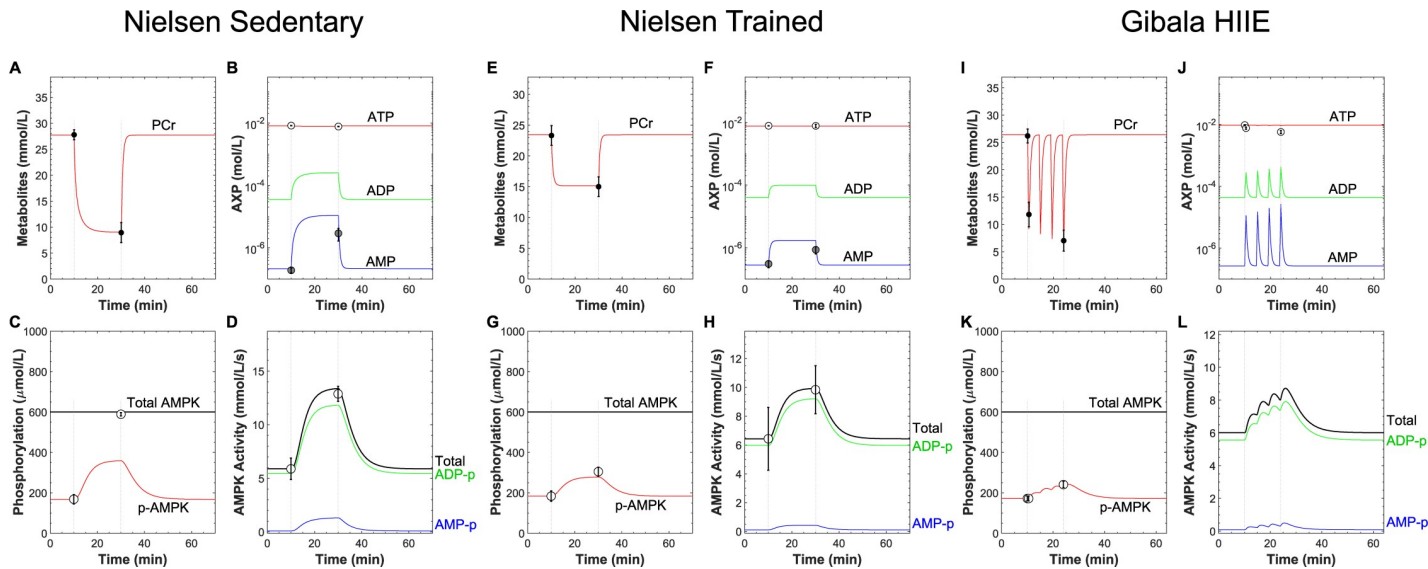

**Fig 3. Model validation.** The model predictions were compared to data collected from continuous cycling exercise performed by sedentary (panels A-D) or trained human participants (panels E-H) [41] and sprint-interval cycling exercise [6] (panels I-L). The hatched vertical lines indicate the times at which exercise was initiated and ceased. Data points and error bars represent the means ± standard error. A, E, I) Time courses of phosphocreatine (PCr) concentration. The filled circles represent the measured concentration. B, F, J) Semi-logarithmic plot showing the time courses of adenine nucleotide concentrations in response to exercise. The open and grey-filled circles represent the measured ATP and AMP concentrations, respectively. C, G, K) Time courses of total and phospho-AMPK (p-AMPK) levels in response to exercise. The open circles represent measured p-AMPK levels. D, H, L) Time courses of the simulated kinase activities for each AXP-p-AMPK complex. The open circles represent the mean relative change in total AMPK activities between rest and exercise.

and transient increases of AMP concentrations (Fig 3B and 3F), phospho-AMPK levels (Fig 3C and 3G), and AMPK activities (Fig 3D and 3H) for the sedentary and trained participants. The quantitative agreement between the model predictions and measured values was satisfactory, except for the change in phospho-AMPK levels during exercise in the sedentary group, which the model underpredicted by approximately 50% (Fig 3C). The model predicted that 89% of the total AMPK activity was attributable to ADP-bound phospho-AMPK.

The Gibala study featured measurements of PCr concentrations and relative phospho-AMPK levels in response to four 30-s sprint intervals interspersed with 4 min of passive recovery. We set the $k_{stim}$ parameter (Table F in S1 Text) to simulate the drop in PCr levels to approximately 12 mM after the first interval (Fig 3A). Thereafter, we increased $k_{stim}$ by ~4% for each subsequent interval to model the progressive depletion of PCr across the four intervals to ~7 mM (Fig 3I). The model predicted virtually constant ATP concentrations throughout the exercise protocol, whereas the measured concentrations noticeably decreased (Fig 3J). Both ADP and AMP levels showed transient increases in response to each interval, with ADP levels increasing from 44 μM to 422 μM and AMP from 0.27 μM to 26 μM by the fourth interval (Fig 3J). The model predicted that the fold-increase in AMP concentration (96-fold) varied approximately as the square of the fold-increase in ADP concentration (9.59-fold). With respect to signaling, the model successfully reproduced the observed ~30% increase in phospho-AMPK levels in response to the four intervals (Fig 3K). The model predicted that AMPK activity increased from 6.0 mM/s to 8.6 mM/s, of which 92% was attributable to the ADP-bound form and 4% to the AMP-bound form (Fig 3L). Collectively, these results suggest that the model satisfactorily captures the behaviors of AMPK signaling in response to diverse exercise protocols.

## Global parameter sensitivity analysis

The above results demonstrate that the model is capable of reproducing AMPK phosphorylation and activity kinetics during exercise and that AMPK activity is mainly determined by ADP-bound complexes. However, the data on which the model is based are insufficient to constrain the uncertainties of the model's numerous parameter values, such that AMP-dominant control could be possible if different parameter values were used. Second, the result provides little insight as to why ADP is predicted to be the dominant controller and how the different mechanisms contribute to overall control.

To address these issues, we performed MPSA. In the unconstrained MPSA, 216 of 50,000 simulations exhibited acceptable phospho-AMPK kinetics, while the α1β2γ1 $K_D$ and α2β2γ3 $K_D$ MPSAs respectively resulted in 2,799 and 2,685 acceptable simulations (Supplementary spreadsheet "MPSA" worksheet). Of the acceptable runs in the unconstrained MPSA, 77 simulations exhibited ADP-dominant control of AMPK activity and 139 exhibited AMP dominance (Supplementary Spreadsheet "MPSA" worksheet). The α1β2γ1 $K_D$ MPSA resulted in 2,717 and 82 simulations featuring ADP- and AMP-dominant control, respectively, while the α2β2γ3 $K_D$ MPSA resulted in 2,281 and 404 simulations featuring ADP- and AMP-dominant control, respectively (Supplementary Spreadsheet "MPSA" worksheet). These results confirmed that AXP-dominance depends on the parameter values.

We next determined the parameters that were most important for determining AXP dominance in each of the three MPSA. We focused on parameter groupings that emerged from Equation I (S1 Text), specifically the relative $K_D$ values of the AXP-(p)-AMPK reactions, the enzyme kinetic parameters of the AMPK kinase and phosphatase ($V_{max}$ and $K_M$), which determine [*p-AMPK*], and the activator concentrations and allosteric activation potencies. In the unconstrained MPSA, AXP dominance was determined mainly by the rate constants for the binding of ADP and AMP to phospho-AMPK, i.e., *k10f*, *k10r*, *k11f*, and *k11r*, as indicated by their higher K-S statistics and lower P-values by comparison to the other parameters (Supplementary Spreadsheet "MPSA" worksheet). Of these, *k11r*, the dissociation rate constant for AMP-p-AMPK, had the highest correlation coefficient with the fraction of ADP control (0.38, Supplementary Spreadsheet "MPSA" worksheet). The effects of these individual parameters propagated into the corresponding $K_D$ values for the two reactions (S1A Fig, top right panel) as well as to ratios of $K_D$ values involving the $K_D$ for the AMP-p-AMPK binding reaction (S1A Fig, bottom panel; note the upward shifts in the boxplots for all $K_D$ ratios featuring "AMP$_p$"). Conversely, K-S statistics, correlations, and boxplots for the $V_{max}$ and $K_M$ parameters for the AMPK kinase and phosphatase were similar between AMP- and ADP-dominant models (Supplementary Spreadsheet "MPSA" worksheet and S1B Fig). ADP and AMP concentrations and AMP-mediated allostery were also similarly distributed for both AMP- and ADP-dominant models (S1C Fig). Overall, the unconstrained MPSA demonstrated that AMP-dominant control of AMPK activity is possible and resulted mainly from parameters determining AXP-p-AMPK binding affinities.

In the α1β2γ1 $K_D$ MPSA, the model was sensitive to the dissociation rate constants for ADP-p-AMPK (*k10r*) and AMP-p-AMPK (*k11r*), as indicated by the K-S statistics and Spearman correlations (Supplementary Spreadsheet "MPSA" worksheet). However, the shifts in the quantiles for the $K_D$ and $K_D$ ratios that incorporated these parameters were relatively minor and not visually different compared to those that did not incorporate the *k10r* and *k11r* parameters (S2A Fig). In contrast, sensitivities were evident for the $V_{max}$ and $K_M$ parameters for the AMPK kinase and phosphatase. The K-S statistics and Spearman correlations for the $V_{max}$ and $K_M$ of the ADP-AMPK kinase ($V_{maxKinaseADP}$ and $K_M16$, respectively) were the highest amongst the enzyme kinetic parameters (Supplementary Spreadsheet "MPSA" worksheet).

These results were corroborated by the boxplots, which additionally showed that the sensitivities propagated into the corresponding $V_{max}/K_M$ ratio for the ADP-AMPK kinase, the quantile values of which were higher in ADP-dominant models compared to AMP-dominant models (S2B Fig). The sensitivities also propagated into the higher-order ratios incorporating this $V_{max}/K_M$ ratio (S2B Fig, bottom panel).

The parameter to which the model was most sensitive in the α1β2γ1 $K_D$ MPSA was $K_{eqADK}$, which is part of the adenylate kinase reaction rate equation (reaction 5, Table B in S1 Text). This parameter strongly associated with the concentration of AMP (S2C Fig), which is one of the determinants in Equation I (S1 Text). The K-S statistic and Spearman correlation coefficient for $K_{eqADK}$ were 0.56 and 0.46, respectively, both of which were the highest observed in the α1β2γ1 $K_D$ MPSA (Supplementary Spreadsheet "MPSA" worksheet). The boxplots revealed that AMP-dominant models were associated with lower medians for $K_{eqADK}$, higher medians for maximum AMP concentration (~7 µM vs. 2 µM), lower medians for ADP/AMP concentration ratios, and higher AMP allostery (~8 vs. 5-fold) (S2C Fig). Collectively, these results show that AMP-dominant control can occur when changes to parameters other than those determining AXP-(p)-AMPK binding affinities are made.

Similar results were observed from the α2β2γ3 $K_D$ MPSA, except that the model was not sensitive to the binding rate constants because their sampling ranges were tightly constrained (S3A Fig). The K-S statistics and Spearman correlations were highest for the $V_{max}$ and $K_M$ of the ADP-AMPK kinase ($V_{maxKinaseADP}$, $K_M16$) and $K_{eqADK}$ (Supplementary Spreadsheet "MPSA" worksheet). These results were corroborated by the boxplots, which showed a prominent upward shift in the $V_{max}/K_M$ of the ADP-AMPK kinase for the ADP-dominant models (S3B Fig, top right panel). The K-S statistics and Spearman correlation coefficients for the $V_{max}$ and $K_M$ of the AMP-AMPK kinase ($V_{maxKinaseAMP}$ and $K_M18$, respectively) were slightly elevated (Supplementary Spreadsheet "MPSA" worksheet), and the boxplots showed a downward shift in the $V_{max}/K_M$ ratio for the ADP-p-AMPK phosphatase (S3B Fig, top right panel). Boxplots revealed lower medians for $K_{eqADK}$, higher medians for maximum AMP concentration (~4.5 µM vs. 2 µM), lower medians for ADP/AMP concentration ratios, and similar AMP allostery (1.4- to 1.5-fold) for the AMP-dominant models (S3C Fig). Collectively, the α2β2γ3 $K_D$ MPSA revealed that AMP-dominant models are possible given the uncertainties in the parameter values, especially given the more pronounced differences in $K_D$ values for AMPK binding to AMP and ADP.

## AMP-dominant models from the α2β2γ3 $K_D$ MPSA are implausible

The results from the α2β2γ3 $K_D$ MPSA indicated that AMP-dominant control of AMPK activity is possible given plausible sampling ranges for the parameter values. However, careful inspection of the $V_{max}/K_M$ ratio plots revealed that these values for the ADP-AMPK kinase were shifted to lower values than those for ATP-AMPK in the AMP-dominant models from both the α1β2γ1 $K_D$ and α2β2γ3 $K_D$ MPSAs (top right panels for S1B Fig, S2B Fig, S3B Fig). This result contradicts data indicating that ADP enhances the rate of AMPK phosphorylation by its kinases compared to ATP [14,18,22], such that these models are implausible. We proceeded to ask how many AMP-dominant models from the α2β2γ3 $K_D$ MPSA featured unrealistic parameter values or otherwise dubious predictions that would invalidate these models, which could lead to a more definitive conclusion regarding AXP-mediated AMPK activity control.

For each acceptable model in the α2β2γ3 $K_D$ MPSA, we evaluated the plausibility of the maximum AMP concentration, the relative values of $V_{max}/K_M$ ratios for the ADP-AMPK and ATP-AMPK kinases and phosphatases, and the relative $V_{max}$ values for the same molecular

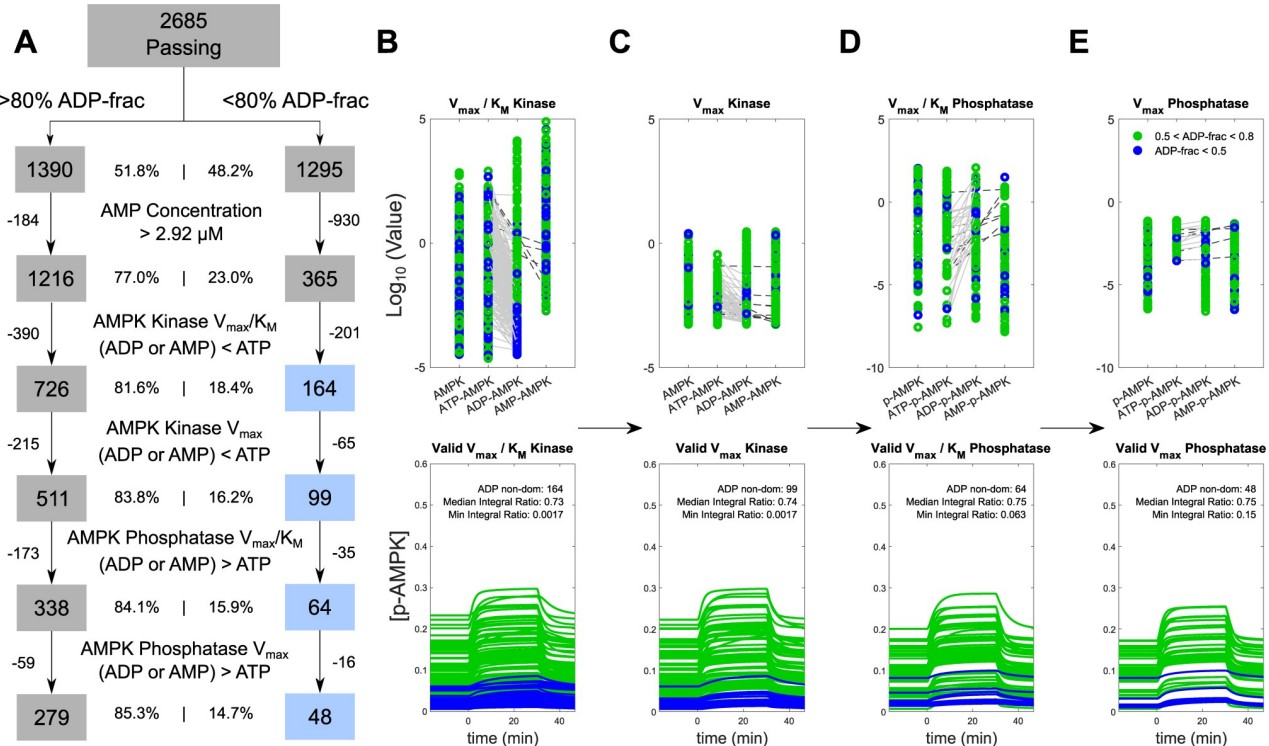

**Fig 4. Evaluation of model plausibilities from the α2β2γ3 K$_D$ MPSA against data-informed constraints.** A) Workflow of the progressive elimination of implausible models. After partitioning out the simulations into ADP-dominant (left boxes, >80% ADP-frac) or ADP non-dominant cases (right boxes, <80% ADP-frac), we then imposed data-informed constraints on the parameter values and ratios. The labels located in between the workflow boxes represent criteria for identifying models as implausible. B-E) The panels from left to right show the progressive elimination of implausible models from the simulations in the α2β2γ3 K$_D$ MPSA in which ADP-p-AMPK contributed less than 0.8 of the time integral of AMPK activity. The panels from left to right correspond to the models featured in the blue-shaded boxes from top to bottom in panel A. Points and curves colored green or blue indicate simulations with *fractions of ADP control* that tend to either ADP- or AMP-dominant control, respectively. The *upper panels* display the log$_{10}$ values of the quantities shown in the top label of each plot for the labelled AMPK complexes below each plot. The green and blue points indicate values from models exhibiting ADP- and AMP-dominant control, respectively. The grey lines indicate implausible relative parameter values between ATP-p-AMPK and ADP-p-AMPK, whereas the black-hatched lines indicate implausible relative parameter values between ATP-p-AMPK and AMP-p-AMPK. which were subsequently eliminated. The *lower panels* represent the corresponding time courses of p-AMPK. The "Median Integral Ratio" and "Minimum Integral Ratio" correspond to the median and minimum values for the *fractions of ADP control* for the ensemble of models. B) First pass, simulations in which the V$_{max}$/K$_M$ values of the ***kinase*** for ADP-AMPK and AMP-AMPK were lower than ATP-AMPK. C) Second pass, simulations in which the V$_{max}$ values of the kinase for ADP-AMPK and AMP-AMPK were lower than those for ATP-AMPK. D) Third pass, simulations in which the V$_{max}$/K$_M$ of the ***phosphatase*** for ADP-p-AMPK and AMP-p-AMPK were higher than ATP-p-AMPK. E) Fourth pass, removing simulations in which the V$_{max}$ of the phosphatase for ADP-p-AMPK and AMP-p-AMPK were higher than ATP-p-AMPK.

species. Of the 2,685 acceptable models, 1,295 featured <0.8 fraction ADP control, thus indicating substantial contribution of AMP to AMPK activity control. Of these 1,295 models, 365 featured plausible free AMP concentrations, i.e., less than 2.92 μM, which is the highest measured value in studies of AMPK signaling in humans in response to moderate-intensity exercise [5,41,46–48] (see S1C Fig, S2C Fig, and S3C Fig, middle panels). However, 201 of these simulations featured V$_{max}$/K$_M$ values for ADP- or AMP-AMPK kinase that were lower than those for ATP-AMPK kinase (Fig 4A and 4B) and were deemed implausible. Of the 164 remaining models, 65 featured ADP- or AMP-AMPK kinase V$_{max}$ values that were lower than those of the ATP-AMPK kinase V$_{max}$ (Fig 4A and 4C). Similarly, the V$_{max}$/K$_M$ values for the ADP- or AMP-p-AMPK phosphatase were higher than those of the ATP-p-AMPK phosphatase in 35 of the 99 remaining models (Fig 4A and 4D). Of the 64 remaining models, 16 featured higher V$_{max}$ values for the ADP- or AMP-p-AMPK phosphatase compared to the

ATP-p-AMPK phosphatase (Fig 4A and 4E). This process of elimination thus left 48 models from the starting 1,295 acceptable models from the α2β2γ3 $K_D$ MPSA. The same process was executed on the 1,390 simulations that featured >0.8 fraction ADP control, of which 279 had appropriate kinetics in the final subset (Fig 4A).

To evaluate the plausibility of the 48 remaining models, we compared the values of the $V_{max}/K_M$ ratios for the ADP-(p)-AMPK and AMP-(p)-AMPK kinases and phosphatases. Experimental data indicates that these parameters are likely different and in manners that favor AMP control; however, they are unlikely to be *substantially* different based on their measured dose response curves [14,22], which we conservatively defined for the analysis as 100-fold. In 29 of the 48 models, we observed that at least one of the $V_{max}$ or $K_M$ parameters was 100-fold different in the direction favoring the creation of AMP-p-AMPK or the removal of ADP-p-AMPK (Supplementary Spreadsheet "48 models" worksheet). In the remaining 19 models, the highest share of AMP-p-AMPK as a contributor to total AMPK activity was 32%, compared to 68% contributed by ADP-p-AMPK.

Finally, we examined the model-predicted time courses of AMP-p-AMPK, ADP-p-AMPK, and total phospho-AMPK from the 48 models to look for patterns that might explain the increased degree of AMP control in these cases. For the AMP-dominant models, the total phospho-AMPK was low by comparison to ADP-dominant models (Fig 4, lower panels, compare blue and green curves). Plots of ADP-p-AMPK time courses revealed very low values (S4 Fig). AMP dominance therefore only seemed possible in models predicting unrealistically low ADP-p-AMPK and total phospho-AMPK concentrations. Overall, the results indicated that the AMP-dominant or shared control models in the α2β2γ3 $K_D$ MPSA were implausible.

## The model generalizes to exogenous direct-binding AMPK activators

Based on the insights gleaned from the above analyses, we wondered whether the model would generalize to additional exogenous direct-binding AMPK activators. If the four determinants of activation potency are adequately represented in the model, then substituting activator-specific parameter values should enable the model to reproduce experimentally measured AMPK activities. Successful prediction of the measured AMPK activation would further validate our model and extend its applicability to understanding AMPK pharmacology.

We first confirmed that the model was able to reproduce observed intramuscular ZMP concentrations over time in response to 5-aminoimidazole-4-carboxamide ribonucleoside (AICAR) perfusion (Fig 5A) [49]. All parameter values relating to ZMP-mediated AMPK activation were obtained from reports of independent measurements and were not otherwise calibrated to the data. The model predicted that ZMP elicited similar levels of phospho-AMPK but higher AMPK activities than those elicited by moderate exercise (Fig 5B and 5C). The model-predicted phospho-AMPK activities corresponded well to experimental measurements (Fig 5C).

Similar to ZMP, we substituted parameter values from independent *in vitro* experiments to simulate the effects of C991. In response to bolus addition of C991, the model predicted increasing levels of phospho-AMPK and AMPK activity as the dose of C991 increased from $10^{-4}$ mM to $10^{-2}$ mM. Each simulation agreed well with the experimental data (Fig 5B and 5C) [50]. The $10^{-2}$ mM dose of C991 elicited lower phospho-AMPK levels but higher AMPK activity, when compared to AICAR and moderate exercise (Fig 5B and 5C). Together, these results suggest that the model generalizes to direct-binding AMPK activators.

Next, we summarized how the properties of the different AMPK activators integrate to produce emergent AMPK activation by simulating the dose response curves of each activator. The model qualitatively reproduced the sigmoidal shape expected for these relationships (Fig 5D).

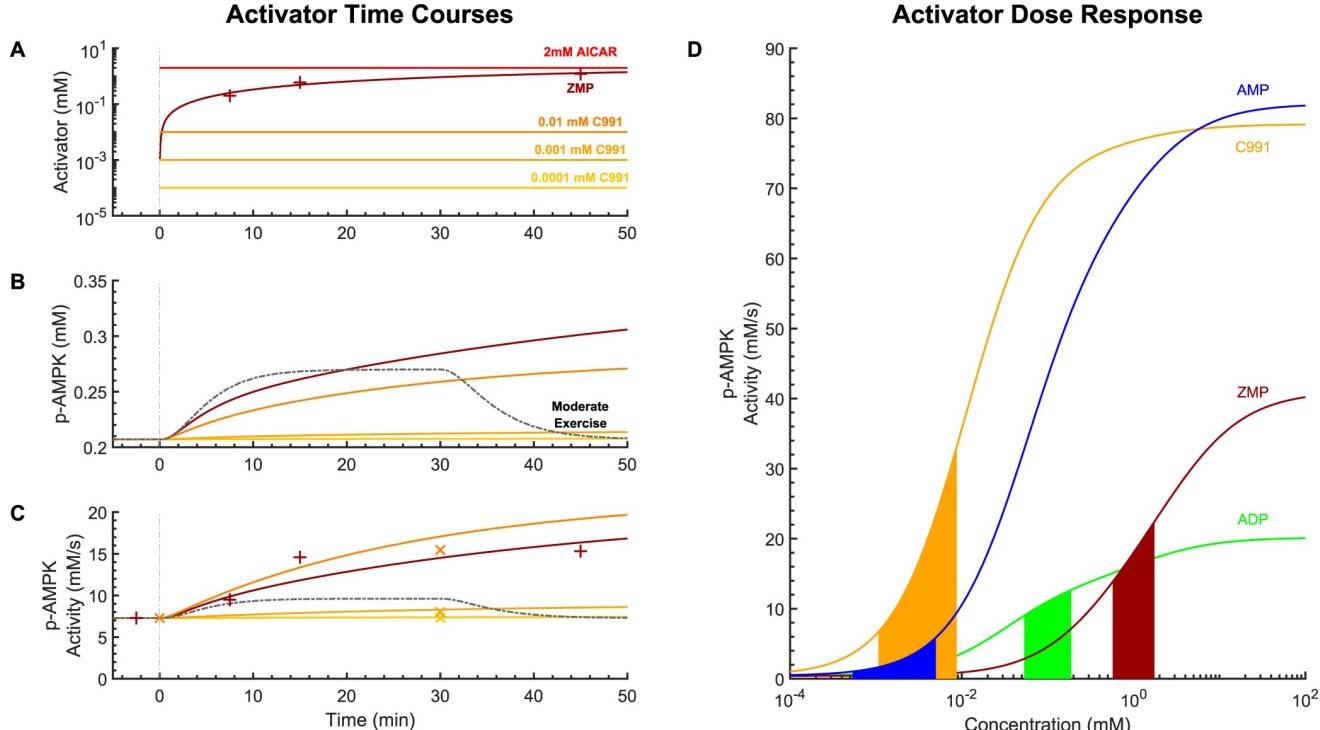

**Fig 5. AMPK pharmacological activator analysis.** Time courses and dose response of simulated AICAR perfusion, simulated Compound 991 incubation, and moderate intensity exercise (63% $\dot{V}O_{2peak}$). A) Time courses of concentrations of activators. To simulate perfusion of AICAR, we assumed that its concentration remained unchanged throughout the simulation. The concentration of ZMP was fitted to published data. B) Time courses of phospho-AMPK (p-AMPK) concentrations for each simulation. C) Total AMPK activities for each simulation. The "×" and the "+" markers denote measured AMPK activities in response to Compound 991 and ZMP, respectively. D) Dose response of AMPK activities in response to treatment with Compound 991, ZMP, ADP, and AMP. The model was simulated independent of the bioenergetic module by removing reactions 1–5 and setting the AXP = 0 except if added exogenously, in which case its total concentration was held constant by setting generation and degradation reaction rates to zero. The reactions were simulated for 240 minutes and the steady-state values plotted. The filled portions under each curve indicate the ranges of activator concentrations that are either measured (AMP, ADP) or experimentally applied (ZMP, Compound 991).

The model also predicted that C991 elicits the highest AMPK activities of the four when applied at its experimentally applied concentrations, followed in order by ZMP, ADP, and AMP (Fig 5D, filled portions of the curves). While AMP has the potential to be the most potent activator of AMPK based on three of four determinants (Equation I in S1 Text), its low concentration *in vivo* causes it to rank last amongst the activators.

## Discussion

In this study, we developed and analyzed a mathematical model of AMPK signaling in human skeletal muscle to systematically investigate AXP-mediated control of AMPK activity dynamics during exercise. The model incorporates the known mechanisms of AXP-mediated AMPK activation in a single unified framework, thus allowing for their study in context of one another. We applied global parameter sensitivity analysis guided by a simple theoretical framework and data-informed constraints to predict that AMPK activity dynamics in skeletal muscle during exercise are determined principally by ADP and not AMP. Furthermore, we showed that our model can predict the effects of other direct-binding AMPK activators, thus demonstrating its usefulness for understanding AMPK pharmacology.

Our conclusion is based on the following interpretations of our results. First, we showed that the calibrated model predicted that ADP-p-AMPK contributed over 90% to the total

AMPK activity during continuous and sprint-interval exercise. While striking, we could not straightforwardly conclude ADP-dominant control because many of the model's parameter values are uncertain. To analyze the model in the context of these uncertainties, we employed global parameter sensitivity analyses in which the predictions from ensembles of models featuring parameter values sampled from plausible ranges were analyzed. The MPSA revealed that AMP-dominant control is possible for many parameter sets. However, introducing data-informed constraints on free AMP concentrations, the relative enhancement of AMPK phosphorylation when bound to ADP compared to ATP, and the relative protection of phospho-AMPK from dephosphorylation when bound to ADP compared to AMP, led to the elimination of all models featuring AMP-dominant control. Of the remaining plausible models, the highest contribution of AMP to AMPK activity control was 32%, compared to 68% for ADP. Collectively, these results imply that ADP is the dominant controller of AMPK activity dynamics in exercise.

Our study contributes significant new understanding to a longstanding debate regarding how AMPK activity is controlled by the adenine nucleotides. AMP has long been considered the principal controller of AMPK activity, which is reflected in the name of the enzyme [51]. However, subsequent structural studies showed that ADP could also control AMPK activity, and some proposed that ADP is the predominant controller of AMPK activity *in vivo* [18,27]. Evidence in support of AMP as the "true" physiological regulator then followed [22,24], and the debate remained unresolved to date. The crux of the debate is whether the stronger per-molecule activation potency of AMP outweighs the two-orders-of-magnitude higher concentration of ADP in cells. Resolving the debate will require accurate estimates of these quantities and accounting for them in a unified framework. Our study presents a new mathematical model of AMPK signaling that explicitly includes these factors and accounts for the full range of corresponding estimates available in the literature. Quantitative models and arguments have been proposed in prior work [27,52] but these were based on steady-state considerations, whereas our model considers dynamics. Existing dynamic models of AMPK signaling [33–36] only considered AMP as the activating nucleotide and were applied to cell types other than skeletal muscle. In addition, unlike these previous studies, ours was the first to consider estimates from the AMPK isoforms and ones from studies featuring results supporting AMP-dominant control, in particular the 13-fold allosteric activation of the α1β2γ1 isoform by AMP reported by Gowans et al. [22]. In this way, our model integrated all pertinent knowledge to evaluate AXP-mediated control of AMPK activity in a systematic, comprehensive, and fair manner. The model can be easily updated as new estimates become available.

Our model can be applied to study AMPK activity dynamics during exercise. Studying signaling dynamics in exercising humans or animals is currently difficult because it relies on biochemical analyses of biopsied muscle. Since only a few biopsies can be ethically sampled from any one individual, these approaches cannot be used to measure the full dynamic profile of AMPK activity in response to a bout of exercise. Mathematical models provide interpolated estimates between the data points, thus inferring the dynamics. The promise of the AMPK model for the proposed application is evident from the simulations of both continuous and sprint-interval exercise (Figs 2 & 3). The latter shows the accumulation of AMPK activity with each interval despite lengthy (4 min) rest periods, which provides clues regarding how sprint-interval exercise can be so potent for inducing fitness-promoting adaptations in skeletal muscle despite the minimal time spent exercising [53].

Our model also provides a general framework for evaluating small-molecule activators of AMPK, which are being studied as therapeutics for muscle and metabolic dysfunction and as part of exercise-mimetic cocktails [10,54]. Typically, AMPK activators have been identified on the basis of binding affinity and enhanced substrate phosphorylation [54,55]. Early AMPK

activators featured properties based on AMP given its greater per-molecule potency of activation relative to other activators such as ADP [56–58]. However, these criteria are insufficient for predicting therapeutic suitability because they do not consider factors such as the molecule's ability to protect AMPK from dephosphorylation, its pharmacokinetics, or the duration of its action once in the cell [54]. Indeed, our results imply that designing a molecule that faithfully mimics AMP's properties would not work well because its concentration would be too low. Indeed, AICAR, the precursor of the AMP mimic ZMP, is applied in experiments at millimolar concentrations, which after uptake and processing leads to intracellular ZMP concentrations that exceed those of AMP by at least two orders of magnitude (Fig 5). We recommend that drug developers assess candidate AMPK activators based on all the factors expressed in Equation G (S1 Text), rather than just a subset as is current practice. More broadly, integrated approaches to drug design are being advanced in the emerging field of quantitative systems pharmacology [59], and our model could find use as the pharmacodynamic portion of future physiologically based pharmacokinetic-pharmacodynamic models of small molecules targeting AMPK.

Our study features four noteworthy limitations. First, AMP is likely still crucial to the function of AMPK *in vivo*, owing to AMP's tight and relatively permanent binding to the γ-subunit nucleotide-binding site 4 of AMPK, the function of which remains to be discovered [12]. Our conclusion regarding ADP-dominant control is restricted to the *dynamic variation of AMPK activity* during exercise. Second, the model incorporates the simplifying assumptions that the cell is a well-stirred tank reactor and does not account for possible stochastic effects, spatial concentration gradients, or restrictions on localization. Accounting for stochastic and spatial effects is unlikely to affect our conclusions because the concentrations of AMPK signaling components are sufficiently high to be deterministically modeled and because AMP and ADP must exist in the same cellular compartments due to their participation in the same biochemical reactions, e.g., the adenylate kinase reaction. Spatially restricted AMP concentration gradients of sufficient magnitude to outcompete ADP as the main controller of AMPK are unlikely to exist.

Third, we assumed that the parameter values of the model were constant within each simulation, despite the knowledge that exercise imposes generalized biophysical and biochemical effects on cells, such as changes in water content [60], temperature [61], and pH [62], which could cause the parameter values to be nonconstant. For these factors to affect our conclusions, the changes to the parameter values would have to disproportionately affect ADP or AMP relative to one another. We contend that such changes are unlikely to be sufficiently substantial to render AMP as the dominant controller of AMPK, given the results of our extensive parameter sensitivity analyses. Furthermore, any such changes would not necessarily favor AMP. For example, changes to intracellular pH differentially affects the thermodynamics of the bioenergetic reactions involving AMP and ADP. The equilibria of the adenylate kinase and creatine kinase reactions are sensitive to pH [63], but $K_{eqADK}$ varies only up to ~20% depending on the $Mg^{2+}$ concentration between pH 6.4 to 7.4 [63]. The creatine kinase reaction is more sensitive to pH, varying up to ~100-fold for the same pH range [63]. This pH sensitivity is explicitly encoded in the model, with $K_{eqCK}$ expressed as $1.77 \times 10^{(9-\text{pH})}$ (Table D; [39]). Because this reaction operates close to equilibrium in vivo [64], decreased intracellular pH is thought to contribute to decreased free ADP concentration [63,65]. The effect of lowered pH on AMP concentration can be inferred by the equilibrium expression for the adenylate kinase reaction:

$$[AMP] = \frac{[ADP]^2 K_{eqADK}}{[ATP]} \qquad (2)$$

Specifically, reduced pH reduces both ADP concentration and $K_{eqCK}$ as explained above, with the equation implying that the fold-change decrease in ADP would translate to AMP decreasing by the square of this fold change. Collectively, these effects would reduce AMP concentration proportionally more than that of ADP, which would favor ADP-dominant control of AMPK activity. Accordingly, the dynamic pH changes observed in working skeletal muscle are unlikely to affect our conclusions.

Finally, our results depend on accurate measurements of AMP and ADP concentrations available to participate in biochemical reactions in cells, which are subject to uncertainties. Notably, disparate estimates exist for the concentrations of *free* and *total* AXP. At rest, most estimates suggest that the *free* AMP:ATP ratio is ~100-fold less than the free ADP:ATP ratio [18,66] whereas estimates of the *total* AMP:ATP ratio is ~10-fold less than the total ADP:ATP ratio [25,67]. Furthermore, the free concentrations of ADP and AMP are not directly measured but are based on calculations that depend on the assumption that the creatine-kinase and adenylate-kinase reactions operate close to equilibrium, whereas total AXP are directly measured biochemically using high-performance liquid chromatography techniques. In principle, estimates from the latter should be given more weight. However, these issues are unlikely to affect our conclusions for two reasons. First, the apparent discrepancies in the measurements of free and total AXP suggest the existence of pools of AXP tightly bound to cellular structures. These bound AXP would remain unavailable to AMPK, in which case only the free concentrations are relevant to its control. Second, HPLC-based measurements of AXP during and after exercise frequently find no change in the levels of AMP between rest and exercise [41,62,67–69], which if true, obviates total AMP as a possible controller of AMPK dynamics during exercise.

In summary, we have developed and analyzed a mathematical model of AMPK signaling in skeletal muscle during exercise. Our analyses demonstrate that ADP rather than AMP dominates the control of AMPK activity dynamics. More generally, activators of AMPK must feature a sufficient combination of per-molecule activation potency, binding affinity, and concentration to be effective *in vivo*. Our model provides a tool for systematically investigating these factors for any direct-binding AMPK modulator.

## Supporting information

**S1 Text. Supplementary Information.** This file contains Supplementary Methods, Supplementary Tables A-G, Interpretation of S1–S3 Figs, and Supplementary References.
(DOCX)

**S1 Spreadsheet. This file contains calculations for the binding and enzyme kinetic parameter values, as well results for the multi-parametric sensitivity analyses and the parameter plausibility analyses.**
(XLSX)

**S1 Code. This folder contains code needed to run the model and generate Figs 2 and 3.**
(ZIP)

**S1 Fig. Visualization of Unconstrained MPSA results. A**. Effects of AXP-(p)-AMPK binding constants on AMP- versus ADP-dominant control of AMPK activity. *i)* Boxplots summarize the individual parameter values representing the forward ($k_f$) and reverse ($k_r$) rate constants. *ii)* Boxplots summarize the dissociation constants ($K_D$) for each AXP-(p)-AMPK reaction, calculated as $k_r/k_f$ from each simulation. *iii)* Boxplots summarize the ratios of all possible pairs of $K_D$ values. **B.** Effects of parameters determining the enhancement of AMPK phosphorylation on AMP- versus ADP-dominant control of AMPK activity in the Unconstrained MPSA. *i)*

Boxplots summarize the Michaelis constants ($K_M$) and maximum enzyme velocities ($V_{max}$) for the AMPK kinase and phosphatase. *ii)* Boxplots summarize the catalytic efficiencies of the AMPK kinase and phosphatase, which was calculated as $V_{max}/K_M$ for each simulation. *iii)* Boxplots summarize the compound ratios of kinase and phosphatase catalytic efficiencies. **C.** Effects of $K_{eqADK}$, AMP and ADP concentrations, and allostery on AMP- versus ADP-dominant control of AMPK activity in the Unconstrained MPSA. Panels from left to right: i) Boxplots summarize the $K_{eqADK}$ values. ii) and iii) Boxplots summarize the maximum predicted concentrations of ADP (ii) and AMP (iii) during simulated exercise. The dashed-grey horizontal lines represent the reported maximum levels of ADP or AMP in moderate-intensity exercise studies. iv) Boxplots summarize the ratios of the maximum concentrations of ADP and AMP during simulated exercise. v) Boxplots summarize the magnitudes of allosteric activation of the AMP-p-AMPK complex.
(PDF)

**S2 Fig. Visualization of α1β2γ1 $K_D$ MPSA results. A.** Effects of AXP-(p)-AMPK binding constants on AMP- versus ADP-dominant control of AMPK activity. *i)* Boxplots summarize the individual parameter values representing the forward ($k_f$) and reverse ($k_r$) rate constants. *ii)* Boxplots summarize the dissociation constants ($K_D$) for each AXP-(p)-AMPK reaction, calculated as $k_r/k_f$ from each simulation. *iii)* Boxplots summarize the ratios of all possible pairs of $K_D$ values. **B.** Effects of parameters determining the enhancement of AMPK phosphorylation on AMP- versus ADP-dominant control of AMPK activity in the α1β2γ1 $K_D$ MPSA. *i)* Boxplots summarize the Michaelis constants ($K_M$) and maximum enzyme velocities ($V_{max}$) for the AMPK kinase and phosphatase. *ii)* Boxplots summarize the catalytic efficiencies of the AMPK kinase and phosphatase, which was calculated as $V_{max}/K_M$ for each simulation. *Bottom panel*: Boxplots summarize the compound ratios of kinase and phosphatase catalytic efficiencies. **C.** Effects of $K_{eqADK}$, AMP and ADP concentrations, and allostery on AMP- versus ADP-dominant control of AMPK activity in the α1β2γ1 $K_D$ MPSA. Panels from left to right: *i)* Boxplots summarize the $K_{eqADK}$ values. *ii)* and *iii)* Boxplots summarize the maximum predicted concentrations of ADP (*ii*) and AMP (*iii*) during simulated exercise. The dashed-grey horizontal lines represent the reported maximum levels of ADP or AMP in moderate-intensity exercise studies. *iv)* Boxplots summarize the ratios of the maximum concentrations of ADP and AMP during simulated exercise. *v)* Boxplots summarize the magnitudes of allosteric activation of the AMP-p-AMPK complex.
(PDF)

**S3 Fig. Visualization of α2β2γ3 $K_D$ MPSA. A**. Effects of AXP-(p)-AMPK binding constants on AMP- versus ADP-dominant control of AMPK activity. *i)* Boxplots summarize the individual parameter values representing the forward ($k_f$) and reverse ($k_r$) rate constants. *ii)* Boxplots summarize the dissociation constants ($K_D$) for each AXP-(p)-AMPK reaction, calculated as $k_r/k_f$ from each simulation. *iii)* Boxplots summarize the ratios of all possible pairs of $K_D$ values. **B.** Effects of parameters determining the enhancement of AMPK phosphorylation on AMP- versus ADP-dominant control of AMPK activity in the α2β2γ3 $K_D$ MPSA. i) Boxplots summarize the Michaelis constants ($K_M$) and maximum enzyme velocities ($V_{max}$) for the AMPK kinase and phosphatase. *ii)* Boxplots summarize the catalytic efficiencies of the AMPK kinase and phosphatase, which was calculated as $V_{max}/K_M$ for each simulation. *iii)* Boxplots summarize the compound ratios of kinase and phosphatase catalytic efficiencies. **C.** Effects of $K_{eqADK}$, AMP and ADP concentrations, and allostery on AMP- versus ADP-dominant control of AMPK activity in the α2β2γ3 MPSA. Panels from left to right: *i)* Boxplots summarize the $K_{eqADK}$ values. *ii)* and *iii)* Boxplots summarize the maximum predicted concentrations of ADP (*ii*) and AMP (*iii*) during simulated exercise. The dashed-grey horizontal lines represent the

reported maximum levels of ADP or AMP in moderate-intensity exercise studies. *iv*) Boxplots summarize the ratios of the maximum concentrations of ADP and AMP during simulated exercise. *v*) Boxplots summarize the magnitudes of allosteric activation of the AMP-p-AMPK complex.
(PDF)

**S4 Fig. Phospho-AMPK in the 48 Non-ADP-dominant models.** Model-predicted time courses of AMP-p-AMPK and ADP-p-AMPK in the 48 models remaining after the elimination process (see Fig 4 of the main text). The blue, green, and black lines represent the levels of AMP-p-AMPK, ADP-p-AMPK and total AMPK, respectively. The inset numbers represent the corresponding fraction of ADP control.
(PDF)

## Acknowledgments

We thank Drs. Scott Dixon, Eldon Emberly and Greg Steinberg for critically proofreading earlier versions of the manuscript.

## Author Contributions

**Conceptualization:** David C. Clarke.

**Formal analysis:** Ian F. Coccimiglio, David C. Clarke.

**Funding acquisition:** David C. Clarke.

**Investigation:** Ian F. Coccimiglio, David C. Clarke.

**Methodology:** Ian F. Coccimiglio, David C. Clarke.

**Project administration:** David C. Clarke.

**Resources:** David C. Clarke.

**Software:** Ian F. Coccimiglio.

**Supervision:** David C. Clarke.

**Validation:** Ian F. Coccimiglio.

**Visualization:** Ian F. Coccimiglio.

**Writing – original draft:** David C. Clarke.

**Writing – review & editing:** Ian F. Coccimiglio, David C. Clarke.

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
