## [Decision Letter · Decision Letter 0]

13 Apr 2020

Dear Dr. Clarke,

Thank you very much for submitting your manuscript "ADP is the dominant controller of AMP-activated protein kinase activity dynamics in skeletal muscle during exercise" for consideration at PLOS Computational Biology.

As with all papers reviewed by the journal, your manuscript was reviewed by members of the editorial board and by several independent reviewers. In light of the reviews (below this email), we would like to invite the resubmission of a significantly-revised version that takes into account the reviewers' comments.

We cannot make any decision about publication until we have seen the revised manuscript and your response to the reviewers' comments. Your revised manuscript is also likely to be sent to reviewers for further evaluation.

Sincerely,

Daniel A Beard

Deputy Editor

PLOS Computational Biology

Daniel Beard

Deputy Editor

PLOS Computational Biology

Reviewer's Responses to Questions

**Comments to the Authors:**

Reviewer #1: The authors present extensive development and application of a new mathematical model to answer the important question of which of the adenine nucleotides AMP or ADP plays the dominant role in exercise-mediated regulation of the metabolic co-ordinator AMPK. Such models are valuable as dynamic AMPK regulation is complex, making delineation of the specific contributions made by activating ligands extremely difficult by cell or physiological measurements. The authors propose their model will aid development of more effective therapeutics targeting AMPK, implicated in onset and progression of a number of prevalent human diseases. The manuscript is well-written, is based on extensive and relevant literature, and employs robust validation of their model to provide confidence in their conclusion that ADP imparts the larger contribution to triggering AMPK activation. Limitations are readily acknowledged and discussed. My comments are focused largely on my expertise in AMPK regulatory mechanisms, in preference to my limited knowledge of mathematical modelling.

I have a couple of comments:

1. Why is consideration not given to AMP clearance mechanisms? e.g. AMP deaminase reaction, which influences the degree to which the AMP/ATP ratio increases during metabolic stress. Plaideau, C., et al (2012). doi: 10.1096/fj.11-198168. I note this is referred to in Fig 1 but not included in the model. Thus changes in [AMP] are likely to be lower than that predicted from the AK reaction alone. Of course, this will not alter the main conclusion of the study, however I consider this to be an important consideration for model refinement and one that is often over-looked by researchers in arguments supporting the AMP-centric hypothesis, or predicting AMP levels from measured ADP/ATP ratio. At the very least this should be included in the Introduction in line 96, and will likely improve accuracy of the model (e.g. predicted vs measured [AMP] fig 2B).

2. An important component of model validity is selection of appropriate kinetic data. For a2b2g3, why did the authors select allostery data from Rajamohan et al ref 26 in preference to Ross et al ref 42 (Table S5)? The former provides data at sub-physiological ATP 20 uM using bacterial-expressed AMPK, whereas ref 42 employed a range of ATP concentrations up to 5 mM and mammalian cell expressed AMPK. It should be noted in the Introduction line 99 that g3 complexes are generally regarded to be poorly AMP-sensitive. Difference in fold activation between the two studies may be small (1.4 vs. 1.54), but will likely have a large influence on outputs.

Minor comments:

3. line 84: AMPK has several known regulatory phosphosites, the authors need to be more specific here. Suggest changing to ”ADP and AMP inhibit AMPK pThr-172 dephosphorylation and promote AMPK Thr-172 phosphorylation by LKB1”.

4. line 84 & 97. It should be mentioned that AMP and ADP stimulate Thr172 phosphorylation by CaMKKB (Oakhill et al 2010 PNAS doi/10.1073/pnas.1009705107 and ref 18) although I do not propose this element be incorporated into the modelling.

5. line 85: include ref 26 (1st to show ADP protects against pT172 dephosphorylation).

6. line 96-8: this sentence could be misinterpreted, AMP does not affect LKB1 activity directly, but instead makes AMPK T172 a better substrate for upstream kinases.

7. line 105-8: consider changing to “…ADP-mediated net increase in pThr172…”to include both phos and dephos mechanisms.

8. lines 111 & 112: this concept was introduced in ref 24, not 18.

9. line 431….”164 simulations…..” should this read “plausible”?

10. line 501: for clarity suggest changing to…”The 10-2 mM dose of C991 elicited lower phospho-AMPK levels but higher AMPK activity, when compared to AICAR and moderate exercise (Fig. 5B and 5C)”. The authors could mention this is consistent with the view that synthetic AMPK agonists elicit robust AMPK signalling without affecting Thr172 phosphorylation (Scott et al 2015, http://dx.doi.org/10.1016/j.chembiol.2014.03.006).

11. ref 26 and 42 are identical.

Reviewer #2: Coccimiglio and Clarke describe a detailed metabolic model that focuses on AMPK, its substrates, products, regulators and activators (physiological and pharmacological). The authors simulate exercise regimens to provide insights into what signals might stimulate use-dependent changes in muscle. The authors conclude that “ADP is the dominant controller of AMPK activity dynamics in skeletal muscle during exercise by virtue of its higher concentration.”

Major

This is a well-written manuscript and the results may have significance for understanding what cellular signals drive muscle plasticity. In addition, there is significance for pharmaceuticals and dietary supplements, “exercise mimetics” that include small molecule activators of AMPK.

My biggest concern about this manuscript is what the authors consider to be the physiological range of free, cytoplasmic [ADP]. This may have a major impact on the authors’ conclusion. The concentration range of AMP specified on pate 5, line 6 from bottom, seems off (specifically too high) at the lower end. For example, see Kushmerick et al., 1992, Mammalian skeletal muscle fibers distinguished by contents of phosphocreatine, ATP, and Pi. Proc. Natl. Acad. Sci. USA 89:7521-7525. There is additional relevant ref data in related papers from the Kushmerick lab, e.g., Chase and Kushmerick, 1995, Effect of physiological ADP levels on contraction of single skinned fibers from rabbit fast and slow muscles. Am. J. Physiol. 268:C480-C489. The authors’ argument that “ADP is the dominant controller“ relies in part on their higher concentrations of ADP relative to AMP (in combination with affinities). My reading of the 31P NMR literature on muscle suggests that the range of ADP indicated by the authors would only be found during heavy exercise or hypoxia. The authors therefore need to clarify what happens at lower ADP levels and whether their conclusion would change; perhaps in a 3D plot define the concentration ranges over which ADP or AMP would dominate.

Minor

Once the authors clarify the physiological range of ADP and its impact on the conclusion (per above), the two following statements could be expanded.

In the concluding line of Abstract, clarify “ADP is the dominant controller of AMPK activity dynamics in skeletal muscle during exercise by virtue of its higher concentration.” Specifically, clarify “higher concentration” relative to what (AMP) as stated more clearly in the author summary.

In the Author Summary: “How the adenine nucleotides interact to control AMPK activity is poorly understood.” Clarify or eliminate because there are plenty of 3D structures and in vitro biochemistry.

Reviewer #3: This paper deals with an important issue, and uses a computational approach to draw some interesting conclusions. I have a few specific comments, and one less so.

I’d like to see a little more detail about the bioenergetic module, which is basically Vicini plus Lambeth (to use a lazy shorthand). Lambeth models glycogenolysis to lactate in a closed system at constant pH (assuming a fixed glycogen phosphorylase a/(a+b) ratio), Vicini models oxidative phosphorylation, and both include the creatine kinase system. The present paper I think assumes no lactate production (i.e. pyruvate is fully oxidised) and although pH-dependence of various model parameters is (rightly) included explicitly, the actual pH is fixed in the model. I have a few comments/requests:

• It would be useful to have this explicitly stated in the Supplement, along with the reasons for the simplifications.

• The ‘The module includes…’ section in the Supplement should presumably contain glycolysis.

• To be clear, this approach seems reasonable to me, and unlikely to affect the novel results which are the point of the paper. The one possible exception is pH: you do discuss this (p. 22) but I’m not sure why you conclude ‘.. thus reinforcing our conclusions’ (line 635).

A more general comment on the physiological context. I agree that ‘Enhanced understanding of AMPK activation would inform both exercise training biology and AMPK pharmacology’. I’m struck by the difference in time-scale between three things: (i) AMPK activation by ADP and AMP (which go up and down, roughly together, in potentially complicated patterns as the myocyte happens to start, stop or change the intensity of contractions; and much of the time are around resting levels), (ii) the downstream consequences, dependent on complex series of gene activations, and (c) AMPK activation by potentially therapeutic pharmacological agents.

• Presumably what relates (i) and (ii) is just the averaging/integrating effect of multiple processes. Does this model have anything to say about that?

• What is the best way of thinking of the relationship of the pharmacology of (iii) to that physiology? Fig 6 simulates direct activation of AMPK by infused activators (alone and together), but how would that interact with the exercise response?

Minor points

• Refs 35 and 39 are the same

• Something about the phrasing of ‘Reduced pH provides a driving force away from ADP and towards ATP in the creatine kinase reaction’ doesn’t seem quite appropriate to an equilibrium. If Cr/PCr is unchanged, yes, a lower pH implies a lower [ADP], but embedded in the whole metabolic regulatory network that condition can’t be assumed.

**Have all data underlying the figures and results presented in the manuscript been provided?**

Reviewer #1: Yes

Reviewer #2: Yes

Reviewer #3: Yes

PLOS authors have the option to publish the peer review history of their article (what does this mean?). If published, this will include your full peer review and any attached files.

Reviewer #1: Yes: Jonathan S Oakhill

Reviewer #2: Yes: P. Bryant Chase

Reviewer #3: No
---

## [Decision Letter · Decision Letter 1]

15 Jun 2020

Dear Dr. Clarke,

Thank you very much for submitting your manuscript "ADP is the dominant controller of AMP-activated protein kinase activity dynamics in skeletal muscle during exercise" for consideration at PLOS Computational Biology. As with all papers reviewed by the journal, your manuscript was reviewed by members of the editorial board and by several independent reviewers. The reviewers appreciated the attention to an important topic. Based on the reviews, we are likely to accept this manuscript for publication, providing that you modify the manuscript according to the review recommendations.

Specifically, in my opinion it would only strengthen the manuscript to follow-up on the remaining suggestions of the reviewers.

Sincerely,

Daniel A Beard

Deputy Editor

PLOS Computational Biology

[LINK]

Reviewer's Responses to Questions

**Comments to the Authors:**

Reviewer #1: The authors have addressed all my concerns except one, which may have been missed:

It should be noted in the Introduction line 99 that skeletal muscle gamma3 complexes have been shown to be poorly AMP-sensitive (in terms of allosteric activation) in studies that directly compare all 3 gamma isoforms eg Ross et al Biochem. J. 473, 189–199 (2016) and Langendorf et al Nat. Commun 7, 10912 (2016).

Reviewer #2: Coccimiglio and Clarke have provided extensive documentation in their response to reviews of their original submission.

Major

In the previous review, I stated that “My biggest concern about this manuscript is what the authors consider to be the physiological range of free, cytoplasmic [ADP].” In their responses, the authors state that “the concentration estimates of the adenine nucleotides are critical to our conclusions” and then state that it is the “concentrations of ADP and AMP relative to each other are the critical driving factor”. The authors then present a simulation with resting ADP = 22.5 uM while their summary table of ADP estimates suggests that resting ADP may be significantly lower than 22.5 uM when we focus on the 31P NMR results. I agree completely that the ratio of ATP/ADP is important for some aspects of metabolism, but the absolute concentrations can also be significant. I don’t understand why they didn’t perform the simulations at lower resting ADP (e.g., 1 and 10 uM) to demonstrate that their conclusions do not change, if that is the case. I hope the authors would agree that good modeling practice is to work not only with parameters that are close to physiological, but in addition, to bracket key values with the range of physiological estimates. Beyond stating “we stand by the [ADP] estimates used in our study and we contend that our conclusions are robust to variations in measured [ADP],” the additional simulations at lower (but according to the 31P NMR data, still physiological) should only strengthen the authors’ conclusions in the eyes of future readers if nothing changes qualitatively.

Reviewer #3: Thank you - you have addressed all my points.

**Have all data underlying the figures and results presented in the manuscript been provided?**

Reviewer #1: None

Reviewer #2: Yes

Reviewer #3: Yes

PLOS authors have the option to publish the peer review history of their article (what does this mean?). If published, this will include your full peer review and any attached files.

Reviewer #1: Yes: Jonathan S. Oakhill

Reviewer #2: Yes: P. Bryant Chase

Reviewer #3: No
---

## [Editor Report · Decision Letter 2]

19 Jun 2020

Dear Dr. Clarke,

We are pleased to inform you that your manuscript 'ADP is the dominant controller of AMP-activated protein kinase activity dynamics in skeletal muscle during exercise' has been provisionally accepted for publication in PLOS Computational Biology.

Best regards,

Daniel A Beard

Deputy Editor

PLOS Computational Biology

Daniel Beard

Deputy Editor

PLOS Computational Biology

---

## [Editor Report · Acceptance letter]

22 Jul 2020

PCOMPBIOL-D-20-00142R2 

ADP is the dominant controller of AMP-activated protein kinase activity dynamics in skeletal muscle during exercise

Dear Dr Clarke,

I am pleased to inform you that your manuscript has been formally accepted for publication in PLOS Computational Biology. Your manuscript is now with our production department and you will be notified of the publication date in due course.

With kind regards,

Laura Mallard
